# Origin, Migration, and Reproduction of Indigenous Domestic Animals with Special Reference to Their Sperm Quality

**DOI:** 10.3390/ani12050657

**Published:** 2022-03-05

**Authors:** Gerhard van der Horst, Liana Maree

**Affiliations:** Comparative Spermatology Laboratory, Department of Medical Bioscience, University of the Western Cape, Bellville 7535, South Africa; lmaree@uwc.ac.za

**Keywords:** indigenous, domestication, sperm quality, sperm traits, natural selection, artificial selection, crossbreeding

## Abstract

**Simple Summary:**

Indigenous domestic animals are derived from “wild” ancestors that have been domesticated as far back as 11,000 BP. In this investigation, we concentrate on indigenous domestic animals such as cattle, sheep, goats, pigs, and chickens and consider their fertility potential. In South Africa alone, more than 60 indigenous domestic breeds have been listed, and by and large, their sperm quality is similar to high fertility exotic breeds. Why are these indigenous breeds important? Particularly during the last 7000 years, different races migrated with their domestic animals, mainly from Northern to Southern Africa, and the animals were exposed to droughts, food scarcity, and many endo- and ecto-parasites. Accordingly, these animals are well-adapted to the harsh conditions of Southern Africa, and it is important to include them in breeding programs to exploit their favorable traits.

**Abstract:**

Indigenous domestic animals such as cattle, sheep, goats, pigs, and chickens have a natural resistance to endo- and ecto-parasites and are tolerant in terms of harsh environmental conditions. These species orginated from the Fertile Cresent between 12,000 and 10,000 BP before migrating into surrounding continents. In view of limited information on the reproductive status of indigenous breeds, it is important to examine their semen characteristics in order to select males to improve livestock production. We have largely relied on existing literature but also our published and ongoing research on sperm quality assessment of several indigenous breeds. The sperm quality of these breeds is similar to current commercial breeds and has been quantified using cutting-edge methods. In this context, we have presented sperm functional tests which provide a better estimate of semen quality than just a standard semen analysis. Initial results suggest that the indigenous breeds have a high sperm quality and sperm functionality similar to currently farmed exotic or crossbreeds. In the long-term, the importance of preserving the favorable traits of these breeds is a priority in view of crossbreeding with existing good meat and milk producers.

## 1. Introduction

Animal domestication in its simplest form refers to a free-ranging or wild species that has been confined by humans for breeding and selection over thousands of years. It is important to note that domestication and the taming of animals are not referring to the same process; taming is a conditioned modification in an animal’s behavior, whereas domestication is a genetic modification of a bred lineage that results in a predisposition toward human association [1].

In this paper, we will be focusing on the reproductive or fertility potential of indigenous domestic male animals. Consequently, female reproduction will not be discussed in detail; however, this does not imply that female reproduction is not equally important. Terminologies such as indigenous, native, and endemic will briefly be visited in the context of domestication. Indigenous species occur naturally and are therefore not accidentally or deliberately introduced but could be occurring in quite a wide geographical area [2]. Furthermore, two subcategories have been devised, namely indigenous unimproved and indigenous improved, where a greater degree of artificial selection has taken place in the latter. A native species is indigenous to a specific region or ecosystem. In contrast, an endemic species is exclusively found in a particular place [3,4,5].

## 2. Historical Perspectives and Supportive Background

By adopting an evolutionary approach and incorporating contrasting views such as natural selection and artificial selection, we deem it necessary to first postulate on the interactions between animals and their environment that lead to the origin of wild progenitors of herbivorous domesticated species. During the Cretaceous Period (derived from the German word Kreide, about 145 million years ago), there seemingly was co-evolution of grass types and grass eaters, which assisted in molding herbivore evolution [6]. The simultaneous evolution of grasslands also played a vital role in the survival of herbivorous dinosaurs. It has only recently been discovered that titanosaur sauropsid dinosaurs (Figure 1), who lived in the late Jurassic and Cretaceous Periods, were the first grass eaters [6]. They established grasslands that also provided ideal feeding grounds for large grass-eating mammals such as mammoths and rhinoceros millions of years later [7]. In Europe, the larger herbivores became extinct during the Neolithic Era with the expansion of agriculture and the domestication of many animals as contributing factors [8]. 

A difficult question that still remains to be answered is what led to the exact process for the start of animal domestication over thousands of years. We know that it began before recorded history, but very little is known as to the exact time and methods employed. It is clear that the ancestors of domesticated animals must have already exhibited traits that made them somehow useful to humans, such as tasty meat, warm coats for clothing, or a natural affinity for people [2].

It is generally accepted that the domestication of many animal species started during the Mesolithic Era (circa 12,000 to 10,500 before the present time (BP), Table 1). Prior to this, wolves were the first animals to be domesticated in the Paleolithic Era, some 35,000 years ago and continuing up to early Mesolithic times, mainly because of a type of symbiosis with humans. A 2017 study found evidence that early dog-like wolves were indeed genetically disposed to be friendly [2]. This largely evolved into a situation of mutual benefits where the dog-like wolves lived off food from humans and were partly protected from predators, while humans might, in turn, have been warned of danger by the wolves/dogs barking [2,7,9]. Our emphasis, however, relates to the domestication of livestock animals, which coincided with a widespread shift from foraging to farming activities among many cultures (Neolithic Era, 11,000–3800 BP) (Table 1).

Despite the uncertainty of the exact dates and mechanisms of very early domestication, it is known where it started in more recent times during the Neolithic Era. The Fertile Crescent, also sometimes referred to as “the cradle of civilization”, was situated in the Near (Middle) East and was known for its fertile soils and are fed by four rivers, namely the Tigris, Euphrates, Jordan, and Nile (Figure 2). Geographically this area, traditionally known as Mesopotamia, and its adjacent areas overlap with parts of Israel, Egypt, Palestine, Turkey, Iraq, Iran, and Syria in modern times. Edible wild crops such as barley and wild wheat were abundant in the Fertile Crescent (these were cultivated in the Neolithic Era) and, together with plentiful grasslands, these areas were populated by herds of many types of mammalian herbivores [1].

Domestication of the mainly herbivorous animals took place sometime between 12,000 to 10,000 BP (Table 2) [5]. One important pre-requisite for domestication was the removal of the alpha male in herds, and so gaining control of the herd [1]. Perhaps more important for early domestication was the selection of species based on traits such as not being aggressive, seeking attention, readily controlled, small homing range, not sensitive to large variations in environmental factors (e.g., temperature), promiscuity, precocious young as well as other factors as outlined by [1,2,5]. Apparently, virtually all present-day domestic animals, including cattle, goats, sheep, pigs, and chickens, had their origins in the Fertile Crescent (Figure 2) in contrast to dogs and cats being domesticated in many different parts of the world.

## 3. Migrations into Africa

This review concentrates on domestic animals in Africa (as an example), and it is conceivable that the various domestic breeds that originated in the Fertile Crescent were similarly relocated either with their human counterparts or on their own to “greener pastures”. Figure 3 shows, as an example, the migratory cattle routes for both *Bos taurus* (humpless) and *Bos indicus* (with a hump, derived from Zebu) from the Fertile Crescent into Africa. These migrations apparently took place as early as 10,000 BP for *B. taurus* and 4000 BP for *B. indicus* to reach the more southern parts of Africa [10,11,12]. During their voyage into Africa, animals were exposed to a multitude of new challenges ranging from droughts, shortage of food, endo- and ecto-parasites, as well as the *Anopheles* mosquito, the vector causing malaria via trypanosomes in both humans and cattle [12]. While *B. taurus* was resistant to malaria, this was less so for *B. indicus*. More recent information indicates that *B. indicus* had some resistance to trypanosomes and was tolerant to dry conditions, heat tolerant, and less prone to tick infections [12].

A further challenge for the southward migration of humans and their livestock involved the tsetse fly (causing sleeping sickness with lethal effects on many body systems, particularly neurological), which formed a barrier just south of the equator. A narrow tsetse-free corridor in the Great Lakes region near Lake Victoria and the Ruwenzori Mountains provided a “Tsetse disease-free pathway” for migration as early as 7000 BP. Natural selection must have been the dominant factor in survival and reproduction in these cases compared to early (and parallel) attempts of humans to apply selection for desired traits (will be discussed in the next section).

Similar migration routes into Africa as indicated for cattle are evident for other domestic animals such as sheep, goats, pigs, and chickens but will not be graphically depicted. There are about 180 breeds of cattle in sub-Saharan Africa, of which 150 are indigenous, and similarly, there are numerous indigenous breeds of sheep, goats, pigs, and chickens in southern Africa [12,13,14,15,16]. Despite this diversity, some of these breeds are endangered mainly due to a range of environmental factors such as droughts and disease.

## 4. Natural Selection vs. Artificial Selection for Reproduction

Early civilizations must have purposely chosen individuals with advantageous traits among livestock, and this signified the start of not only domestication but also artificial selection. However, the earliest domestication efforts would have been supported by events of natural selection in order to adapt to the local and changing agro-environments during cultural migrations [12].

Whereas both natural and artificial selection are involved in the breeding of organisms, there are major differences in their selective pressures and outcomes. Natural selection is an evolutionary force that selects for favorable traits in organisms (postzygotic selection) whereby they become better adapted to their environment for survival and thus produce most of the offspring (“survival of the fittest”) [17]. In contrast, artificial selection is affected by humans and involves the selective breeding of organisms (prezygotic selection) to produce offspring with desirable traits that will be inherited in successive generations [1]. However, in both cases, reproductive success is crucial in order for heritable traits to be passed on to the next generation [9]. It is of no use for animals to have excellent traits but are poor in leaving offspring or if offspring easily succumb, particularly before they have the ability to reproduce.

Interestingly, Charles Darwin considered two types of artificial selection to explain the mechanisms of evolutionary change, namely methodical and unconscious selection [18]. Unconscious selection is the process whereby humans naturally preserve the most valued and cull less valued individuals without any intention to alter the breed. Methodical selection (also called deliberate, conscious, or intentional selection) is probably closest to what is exercised during selective breeding in modern animal husbandry practices, whereby man systematically attempts to modify a breed according to a predetermined standard [9].

Albeit their abovementioned differences, both natural and artificial selection processes require heritable genetic and phenotypic variation among individuals in a population in order for selection to take place [9]. Mutations, selective breeding, and adaptation have differentially influenced the diversity of livestock populations, resulting in modifications in the physiology, morphology, and behavior of domesticated animals [1,19]. Modern breeding systems practiced during the last 300 years intensified the selection of breeding animals in order to meet the demand of human societies in terms of yield in both animal production and revenue.

Breeding characteristics preferred for domestication, such as polygamy or promiscuous mating, males dominating over females, and males initiating the mating process [1], are beneficial for reproductive success under captive breeding conditions. From studies in evolutionary biology, it is well-known that postcopulatory sexual selection in the form of sperm competition typically occurs in polygamous or promiscuous species [20]. Increased levels of sperm competition have been associated with higher relative testis mass, increased semen volume and numbers, as well as a higher percentage of sperm with normal morphology, progressive motility, and viability compared to monogamous species [21]. Thus, male animals of species initially used for domestication most probably had all these favorable semen and sperm characteristics that made them excellent breeders. Even though sexual selection would have played an ever-decreasing role in semen characteristics under domestication [9], males with a high fertility potential, as well as their progeny, were incessantly and successfully selected as breeders for the propagation of desirable traits under controlled practices.

## 5. Advantages and Use of Indigenous Domestic Animals

Artificial selection during domestication, unfortunately, leads to trade-offs between desirable traits and traits not selected for, as is the case with any form of adaptation [9]. For instance, in the case of livestock, selection for more rapid growth has drawn resources away from features such as brain size and acuity of sense organs, which are less important under captive conditions [22]. Similarly, intensive selection of valuable animals for ecological, aesthetical, and economic reasons has resulted in breeds or individuals that are often not able to adapt rapidly sufficiently to changes in the environment resulting from human activities or climate change [11,15]. One possible way to address such lack of or slow adjustment to environmental changes is to switch to species or breeds that are better adapted to the eminent changes or to use them to crossbreed with existing purebred animals.

In the Southern African region, where a hotter and drier climate is imminent, indigenous animal species are usually more heat- and drought tolerant (compared to exotic species), are disease- and parasite resistant and have the ability to utilize poor quality forages and crop residues [12,15,23]. Of the 150 breeds of indigenous cattle in Africa, 25% are found in Southern Africa and derived from the Sanga cattle (*B. taurus africanus*), including breeds such as Nguni, Tuli, Barotse Tswana, Tonga, and Mashona [12]. A good example is the Nguni breed, which is heat tolerant, can survive with minimal water and feeding resources, and is also resistant to several types of tick infections, nematodes, and trypanosomes. While coat color of several domestic species relates to adaptation to specific environments (lighter color in hot environments) as a result of earlier influences of natural selection, there has also been a strong cultural influence for color selection through artificial selection. The great diversity in coat colors and patterns observed among the Nguni cattle of the Zulu people represents a typical example [24].

Since reproduction is influenced by several environmental factors such as temperature, overall animal health, and nutrition, these factors should be considered for optimal conception rates and to meet production goals [25]. Due to both their abovementioned advantageous features and fertility, indigenous livestock breeds have been used in commercial crossbreeding programs to either increase the adaptability of exotic breeds or to improve the efficacy of meat production in indigenous breeds [23].

Unfortunately, only limited information is available on the reproductive status of indigenous livestock in Southern Africa, and this review aims to highlight the advances as well as limitations in male fertility assessments. Fertility is generally defined as the ability to conceive offspring; however, an individual’s or breeding pair’s fertility depends on many factors, including normal development of their reproductive system and gametogenesis, as well as successful fertilization, uterine attachment, embryogenesis, and fetal development [26,27]. While female fertility has received lots of attention and has been enhanced through assisted reproductive technologies and genetic selection, similar efforts in male domestic animals are lacking [25]. Male animals are often the most neglected individuals in a herd, and the implementation of proper screening of semen and sperm characteristics could provide more advanced selection tools to improve livestock production.

## 6. Which Semen and Sperm Traits Have Been Recorded?

The semen and sperm traits of most mammalian domestic species can typically be categorized in three groupings, namely: macroscopic semen traits, e.g., color, volume, and pH; microscopic traits, e.g., percentage sperm motility and sperm concentration; and functional traits, e.g., percentage rapid progressive sperm and percentage hyperactive sperm. In this review, we have incorporated results from existing literature as well as data from our own published and ongoing research on sperm quality assessment of several indigenous breeds (Figure 4) to compare semen and sperm traits of different indigenous and exotic breeds of various domestic species.

The traits for different chicken breeds are compared in Table 3, and these mostly relate to macroscopic semen parameters and some microscopic sperm parameters. What do the data contained in Table 3 [28,29,30,31] indicate in relation to indigenous versus exotic breeds? Independent of country, it seems that Nigerian, Indian, and South African indigenous breeds show similar characteristics for semen volume, semen pH, and the percentage sperm motility. Unfortunately, little information is available on sperm functionality in most chicken breeds; however, several attempts have been made to quantify more detailed sperm motility characteristics and sperm kinematics (see Venda and White Leghorn breeds in Table 3). Overall, it seems that there are great similarities among chicken breeds for macroscopic characteristics worldwide and also for the improved indigenous breeds of South Africa (e.g., Potchefstroom Koekoek). 

Table 4 and Table 5 include macroscopic, microscopic, and some sperm functional characteristics of various indigenous versus exotic breeds of pig, cattle, ram, and goat [32,33,34,35,36,37,38,39,40]. By and large, there are many similarities when several sperm parameters of these breeds are compared for a given species. For example, in Kolbroek boars, compared to Large White boars, most sperm parameters such as percentage motility and several kinematic parameters such as swimming speed (curvilinear velocity or VCL) show agreement (Table 4). Nguni bulls compared to improved indigenous Bonsmara bulls show similarities for pH, semen volume, sperm motility, and sperm concentration. In contrast, there are apparent differences in some sperm kinematic parameters and sperm functionalities, such as the percentage of rapid sperm for these two indigenous breeds (Table 4).

Similar to bulls, the semen parameters for four Southern African sheep breeds and one from Brazil (Santa Inez) show that there are many similarities for semen volume, sperm concentration, percentage motility, and the percentage of live sperm for the indigenous rams (Table 5). Furthermore, there are large overlaps in the different categories of kinematic parameters of St Inez and Merino, but for many sperm functionalities, such as the rapid kinematic categories’ information is limited to one breed and accordingly makes comparisons difficult. However, in goats, comparative data is available for the more important functional kinematic parameters, and there are major agreements for most of the mentioned kinematic parameters among these three goat breeds (Table 5).

For a more in-depth analysis of sperm functional parameters, we compared three South African indigenous bull breeds (Nguni, Bonsmara, and Afrikaner) with three crossbreeds (Simmental × Nguni; Simmental × Bonsmara; Simmental × Afrikaner) using computer-aided sperm analysis (CASA) analysis at 169 frames per second (Table 6). For all three indigenous breeds and at least two crossbreeds, remarkably similar data have been documented for parameters such as percentage total motility, percentage rapid progressive sperm, and most kinematic parameters (general averages and rapid averages). In all the indigenous breeds and crossbreeds, percentage hyperactivation (HA) has been recorded, and except for the Simmental × Bonsmara cross, HA percentages were all above 20% (relates to fertilization success) and was as high as 76% (Siemental × Nguni cross). Isnaini and Wahjuningsih (2019) found that Ongole crossbreed and Simmental bulls had similar semen and sperm quality, but higher sperm concentrations were recorded for Simmental bulls as well as more ejaculates meeting required quality [34].

There is some controversy relating to the reproductive fitness of crossbreds and the potential value to breed for specific traits like improved body mass or milk production. Gregory (2009) indicated that a lack of variability might make organisms less adaptable to a specific environment, or it can cause deleterious mutations to accumulate [9]. Accordingly, it needs to be established what the trade-offs are of crossbreeding. For example, do all the high-quality semen parameters translate to good breeding soundness (see discussion in next section), or can it be used as a yardstick of not only breeding soundness but also the ability to leave fertile offspring with the desired characteristics that have been selected for? Moreover, artificial selection continues to provide an understanding of the basic principles of evolution and natural selection [9]. A recent review of the etiology of subfertility/infertility in crossbred bulls has highlighted that semen production and sperm quality is lower in crossbred bulls compared to indigenous or purebred bulls [41]. The difference in sperm quality is ascribed to lower percentages viability, membrane integrity, and well acrosome integrity, which could be related to the downregulation of sperm protein genes [41,42]. In addition, similar studies involving more sperm functional markers (proteomics and metabolomics) may assist in providing better sensitivity in terms of differences in sperm functionality among indigenous-, exotic- and crossbreeds. For example, Goss et al. (2021) and Alipour et al. (2021) have shown the value of this approach in human sperm [43,44].

## 7. Do Semen Parameters Assessed during Breeding Soundness Relate to Fertility?

Domestic males selected may have good semen quality, and there may well be a pre-selection to breed for specific traits, but what is the status of their breeding soundness? Breeding soundness in bulls encompasses several important components, namely semen quality (at least 30% motility and 70% normal sperm), scrotal circumference (34 cm at 24 months of age), libido, social dominance, as well the ability and speed of mounting the cow. The combination of these is not only important to preserve good fertility in the artificial insemination (AI) industry but is crucial for and affects the reproductive fertility of herd bulls. Furthermore, a single breeding soundness test is not sufficient as this needs to be monitored on a regular basis [45].

What about breeding soundness in other species of domestic animals? Tibary et al. (2018) have alluded to the significance of breeding soundness evaluation (BSE) in rams and bucks and states that it should include a physical examination, an inspection of the reproductive organs, and a semen collection and evaluation [46]. For example, in rams, the minimum scrotal circumference should be 34 cm at 18 months of age, at least 30% of sperm should show progressive motility, and more than 70% of sperm should be morphologically normal. In goats (bucks), sperm motility above 50% and normal sperm morphology above 90% are regarded as excellent [46].

However, a shortcoming in this respect is that most of the macro- and many of the micro-scopic semen/sperm parameters assessed as part of a BSE inform us about the animal’s sperm quality and health status but not necessarily the ability of the animal’s sperm to fertilize an oocyte. Thus, these parameters cannot predict the fertility potential of ejaculates, and additional functional tests should be investigated [47]. Several sperm functions are required by spermatozoa to reach and ultimately fertilize the oocyte and have been related to live birth outcome, including progressive sperm motility, viability, and vitality [48,49].

For example, hyperactivation (HA) takes place in the female reproductive system during the final stages of sperm capacitation (preparing for fertilization). Hyperactive sperm swim at speeds two to three times that in semen, show very large undulations of the head, and portray linearity below 50%. A detailed study of Tankwa goat sperm showed specific sperm kinematic cut-off values for defining hyperactivation and also related this to sperm quality [40]. If the percentage HA of a semen sample exceeds 20%, it is indicative of high functional competence and potential to fertilize the oocyte [50]. Based on the Tankwa goat hyperactivation data and the HA data from Table 6, it is conceivable that most of the bull breeds are indeed potentially fertile. HA data for Tankwa goats [40], as well as indicated in Table 5, further support this notion, and it is conceivable that this may apply to other goat breeds and other domestic species.

Petrunkina et al. (2007) also suggested the development of the capacitation continuum based on the ejaculate’s capacitation response and, in principle, measures ideal capacitation time/status [51]. Three potentially valuable tests suggested by these authors are “(a) the ability to regulate cellular volume in the face of changing environments, (b) the ability to bind to the oviductal epithelium, and (c) the ability to undergo capacitation” in a timely and appropriate manner [51]. In our laboratory, we have developed applicable sperm functional techniques for domestic animals that have been used in the human clinical situation. Among these tests for human sperm are ROS, DNA fragmentation, MMP, hyperactivation [52], and sperm chromatin maturity [53]. In the overall decision of which of these sperm functional tests should be employed, it is important to consider what is practical and feasible for the field situation.

## 8. Is It Important to Protect Indigenous Domestic Breeds?

It is obvious that the true indigenous African domestic breeds represent a large genetic diversity with traits that is important for their survival in harsh climates of Africa and needs to be harnessed and protected. All these aspects make them ideal for breeding in rural communities; however, this diversity is under enormous threat [13,14,15]. The pure Nguni cattle breed, which has been particularly well studied, serves as a good example of a drastic decline in numbers from about 1.8 million in 1992 to less than 10,000 in 2003 [13]. A potential problem relates to crossbreeding programs that select on the one hand for the “survival genes” of the pure indigenous breeds and, on the other hand, select for the larger body mass (mainly muscle mass, beef production) of exotic breeds. Such crossbreeding may also result in animals with poor traits due to unfavorable mutations, as alluded to earlier in the review. A more coordinated program has been suggested to design conservation programs that are aimed at the principal stakeholders, who are mainly local people in the rural areas [13,15].

This latter aspect is particularly important in terms of economic value for the smallholder farmers who keep beef cattle for multiple purposes. Rural farm breeders constitute more than 45% of the total number of breeders globally and even more so for Africa. Meat, milk, cheese, and meat remain important food sources for people in rural communities and even bred in small numbers and without much artificial selection is an important cornerstone for survival and sustainability of rural communities. Moreover, rural households, often representing poor people, depend on domestic animals as an inflation-free form of banking to trading animal products for multiple household expenses [54,55]. It is accordingly important to ensure that the correct scientific-based breeding programs are developed, which is economically viable and sustainable [13,15].

Breeding of domestic species in geographical areas in which it was not possible to breed with exotic species in the past has recently been well-exploited in some parts of South Africa. The Klein Karoo is an arid area in South Africa where sheep breeding is the dominant form of farming. However, breeds such as the Nguni have now been introduced to some parts of the Klein Karoo with great success and are a good example of exploiting their heat-tolerant genetic makeup, among others.

While it is not in the scope of this review to discuss the detailed aspects of cryopreserving the gametes and embryos of domestic animals with desirable traits, it is sufficed to state that it is an important aspect to be considered and also needs to be further developed and applied to indigenous domestic animals.

One question to be explored is how well do their gametes freeze once good genetic material has been identified? This varies greatly, and it is well-known that bull sperm cryopreservation with associated AI is highly successful, while the same cannot be said for rams and goats. In endangered Markhor goats, good post-thaw motility (41% to 53%) is obtained, but fertility trials lack correlation with post-thaw sperm characteristics [56]. It has been highlighted that cryopreservation of sperm must be pursued as a priority, and after all, semen is still the “cheapest” component of livestock breeding [57].

## 9. Conclusions

When farming with indigenous domestic animals, there seem to be two approaches that are important, namely, on the one hand, good scientific farming practices to ensure that the pure breeds are optimally managed and that all their valuable genes are protected (including the long-term cryopreservation of sperm, oocytes, and embryos). On the other hand, good concerted scientific research needs to be performed to find the best sustainable way to breed and farm with crossbreeds. There are furthermore centers in South Africa, Tanzania, and Kenya, among others, who are involved in extensive scientific research on crossbreeding [58]. Data presented in this review seems to confirm that the semen characteristics and many of the sperm functional parameters of the indigenous domestic animals are quite similar to both purebred exotic and crossbreeds and apparently of high quality. This needs to be followed up with breeding soundness examinations. In rural areas, a herd bull will be more often used to service cows than artificial insemination, and it is therefore essential that breeding results in bulls with good semen quality, good ability to successfully mate, and leave fertile offspring adapted to a given environment. It has been emphasized in the context of BSE that semen/sperm quality needs to be extended to further tests on sperm functionality. The Darwinian principle survives that clever artificial breeding taking the above factors into account will largely be a form of natural selection.

## Figures and Tables

**Figure 1 animals-12-00657-f001:**
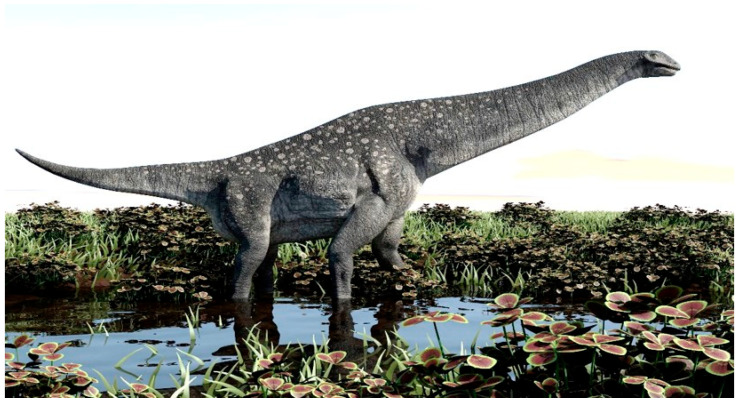
Depiction of a titanosaur sauropsid dinosaur that roamed the earth during the Jurassic and Cretaceous Periods. This grass-eater probably played an important role in the evolution of grasslands and setting the scene for subsequent large herbivores.

**Figure 2 animals-12-00657-f002:**
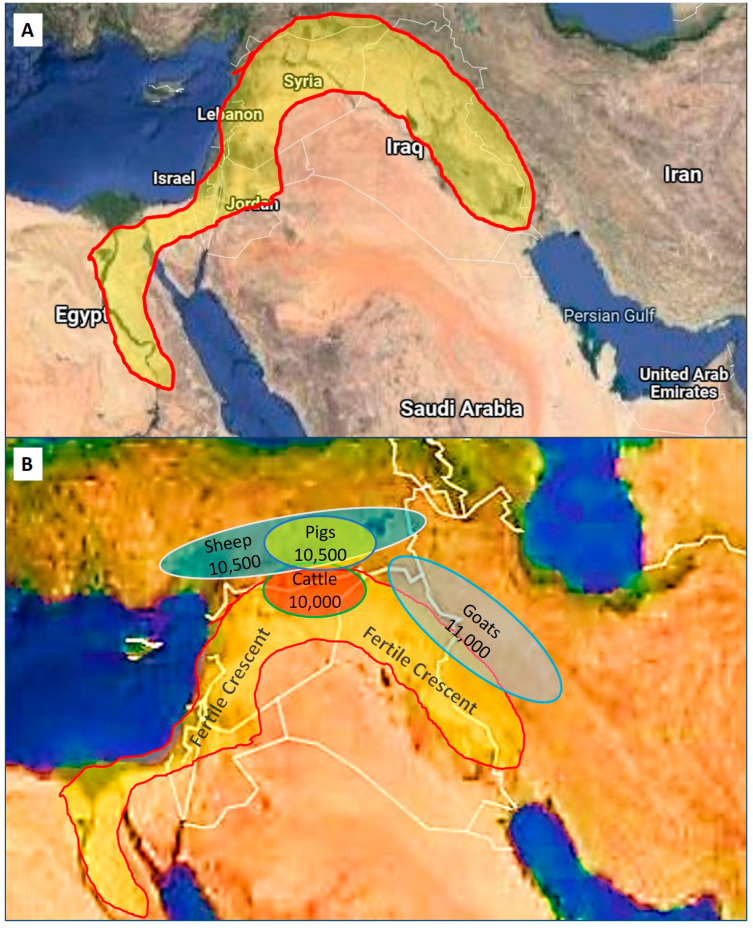
The Fertile Crescent and the origins of domestication. (**A**) The yellow shaded area indicates the size and location of the Fertile Crescent in an area known as Mesopotamia, which overlaps in modern times with parts of Egypt, Israel, Palestine, Syria, Turkey, Iraq, and Iran. (**B**) Shaded ovals represent the approximate areas of domestication of sheep, pigs, cattle, and goats in relation to the Fertile Crescent (after Ref. [1]). Numbers indicate the dates of initial domestication in years before the present (BP).

**Figure 3 animals-12-00657-f003:**
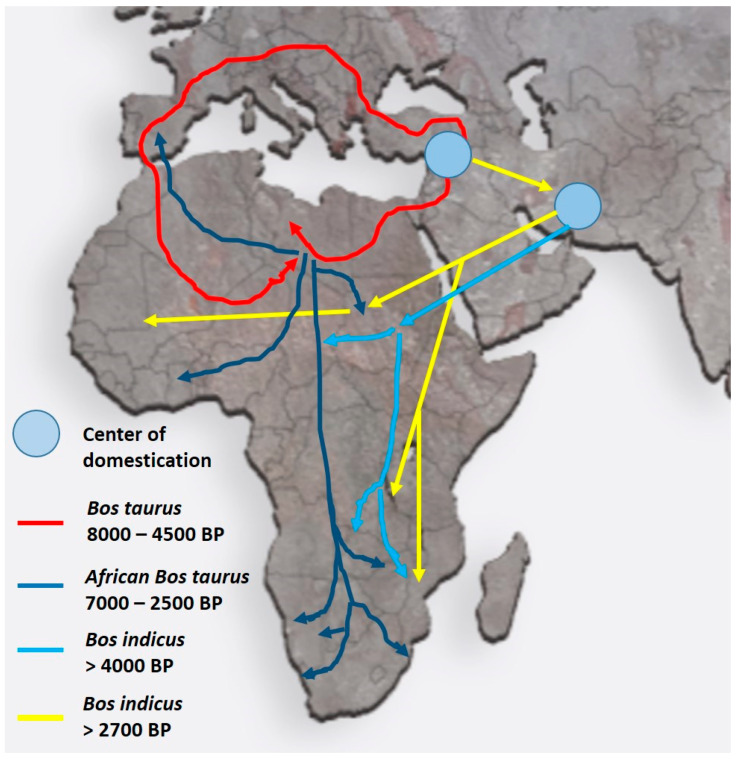
Migratory routes of domestic cattle from their origin in the Fertile Crescent into the northern and southern parts of Africa. Despite having different centers of domestication, both *Bos taurus* and *Bos indicus* breeds migrated into Africa. Colored arrows indicate the different routes and times of migration (after Ref. [12]). The routes and times for the migration of other domestic animals such as goats and pigs are very similar.

**Figure 4 animals-12-00657-f004:**
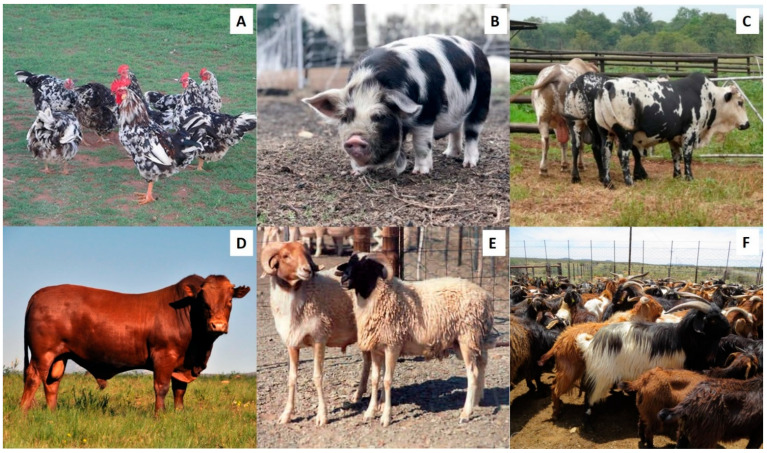
Examples of indigenous domestic breeds in South Africa. (**A**) Venda chickens, (**B**) Kolbroek pig, (**C**) Nguni cattle, (**D**) Bonsmara bull, (**E**) Namaqua-Afrikamer sheep, and (**F**) Tankwa goats.

**Table 1 animals-12-00657-t001:** Development of humans from hunter-gatherers to farming communities during the past 70,000 years. The domestication of animals only started during the late Mesolithic Era, and most present-day livestock species originated during the Neolithic Era.

	Palaeolithic Era	Mesolithic Era	Neolithic Era
	Circa 70,000–12,200 BP	Circa 12,000–10,500 BP	11,000–3800 BP
**Methods of obtaining food**	Hunting	Hunting	Raising and herding animals
	Gathering fruits and grains	Gathering plants and storing food	Farming in permanent villages
**Domestication of animals**	Symbiosis of humans and wolves	First attempts at domestication and taming of “wild” animals	Domestication of most livestock species

**Table 2 animals-12-00657-t002:** Development of humans from hunter-gatherers to farming communities during the past 70,000 years. The domestication of animals only started during the late Mesolithic Era, and most present-day livestock species originated during the Neolithic Era. Data extracted from Zeder (2008) [5].

Domestic Animal	Scientific Name	Domesticated BP	Geographical Area
Dog	*Canis familiaris*	36,000–15,000	Mainly Eurasia
Goat	*Capra hircus*	11,000	Middle East
Sheep	*Ovis aries*	10,500	Middle East
Pig	*Sus domesticus*	10,500	Middle East
Cattle	*Bos taurus/Bos indicus*	10,000	Middle East
Cat	*Felis silvestris catus*	9500	Middle East
Horse	*Equus caballus*	5500	Asia
Chicken	*Gallus gallus domesticus*	4000	Asia

**Table 3 animals-12-00657-t003:** Comparison of semen and sperm characteristics of indigenous and exotic chicken breeds in Nigeria, India, and South Africa. The data presented is focused on Africa but emphasizes that globally there seem to be great similarities between semen and sperm characteristics of indigenous compared to exotic breeds. In instances where a parameter is only presented for one breed, these values are included for future reference.

	Nigeria [28]		India [29]		South Africa [30,31]				
	Indigenous	Amo	Aseel	Kadaknath	Naked Neck	Ovambo	P Koekoek	Venda	W Leghorn *
**Semen and Sperm Traits**	*(n* = 9)	*(n* = 9)	*(n* = 10)	*(n* = 10)	*(n* = 8)	*(n* = 8)	*(n* = 8)	*(n* = 8)	*(n* = 6)
*Macroscopic*									
Semen colour (Scale 1–4)	2.85 ± 0.07	2.0 ± 0.9	Creamy white	Creamy white	−	−	−	−	−
Semen pH	7.0 ± 0.0	7.0 ± 0.0	7.1	7.2	−	−	−	−	6.9
Semen volume (mL)	0.21 ± 0.17	0.2 ± 0.2	−	−	0.3	0.4	0.7	0.2	0.2
*Microscopic*									
Mass motility (Score 1–5)	4.85 ± 0.27	4.37 ± 0.19	−	−	−	−	−	−	−
Total motility (%)	−	−	75.8 ± 7.99	75.7 ± 4.1	65 ± 1.3	80 ± 9.4	90 ± 9.4	96 ± 5.3	80 ± 9.4
Progressive motility (%)	95.0 ± 0.43	82.0 ± 1.15	−	−	−	−	−	−	−
Sperm concentration (×10^9^/mL)	4.93 ± 1.84	3.4 ± 1.07	−	−	1.3 ± 0.01	2.1 ± 0.1	3.5 ± 0.04	2.1 ± 0.45	5.61
Live sperm (%)	88.9 ± 0.58	72.7 ± 0.4	78.4 ± 5.37	82.0 ± 4.47	54.6 ± 0.6	72.9 ± 0.9	88.4 ± 0.7	50.3	56
Normal morphology (%)	81.0 ± 0.78	66.2 ± 0.1	90.0 ± 3.5	93.0 ± 2.8	−	−	−	−	−
*Kinematics*									
Average VCL (µm/s)	−	−	−	−	−	−	−	125.8	55
Average VSL (µm/s)	−	−	−	−	−	−	−	75.5	23
Average VAP (µm/s)	−	−	−	−	−	−	−	93.4	39

* Exotic breed farmed in South Africa. VCL: curvilinear velocity; VSL: straight-line velocity; VAP: average path velocity.

**Table 4 animals-12-00657-t004:** Comparison of semen and sperm characteristics of indigenous and crossbred pigs and cattle.

	Boar [32]		Bull [33,34]		
	SA Kolbroek	SA Large White	Nguni	Bonsmara	Ongole Crossbr
**Semen and Sperm Traits**	*n* = 4	*n* = 4	*n* = 4	*n* = 4	*n* = 1
*Macroscopic*					
Semen pH	7.0 ± 0.0	7.0 ± 0.0	7.1 ± 0.2	7.4 ± 0.3	6.3
Semen volume (mL)	140.4 ± 48.6	177.5 ± 60.4	3.7 ± 1.8	4.5 ± 0.8	6.4
*Microscopic*					
Total motility (%)	95.2 ± 4.2	91.4 ± 6.2	93.0 ± 1.7	91.3 ± 4.2	70.6
Progressive motility (%)	36.8 ± 15.2	22.8 ± 6.8	48.6 ± 12.4	35.3 ± 15.5	−
Rapid motility (%)	55.2 ± 17.3	42.0 ± 13.4	83.6 ± 7.1	54.7 ± 22.2	−
Sperm concentration (×10^9^/mL)	0.72 ± 3.4	0.76 ± 3.7	4.95 ± 1.89	4.88 ± 4.2	8.5
Live sperm (%)	84.6 ± 6.1	81.7 ± 7.1	92.9 ± 4.2	92.7 ± 4.6	−
Normal morphology (%)	−	−	92.1 ± 4.4	89.4 ± 4.1	−
*Kinematics*					
Average VCL (µm/s)	143.3 ± 22.8	129.1 ± 24.8	168.0 ± 20.4	128.3 ± 27.8	−
Average VSL (µm/s)	40.9 ± 10.6	32.8 ± 7.0	76.6 ± 16.5	66.7 ± 18.1	−
Average VAP (µm/s)	90.6 ± 16.1	81.1 ± 17.7	113.7 ± 14.7	53.0 ± 23.8	−
Average LIN (%)	29.2 ± 9.7	25.7 ± 5.1	45.5 ± 7.5	52.7 ± 12.6	−

VCL: curvilinear velocity; VSL: straight-line velocity; VAP: average path velocity; LIN: linearity.

**Table 5 animals-12-00657-t005:** Comparison of semen and sperm characteristics of indigenous and exotic ram and goat breeds. In instances where a parameter is only presented for one breed, these values are included for future reference.

	Ram [35,36,37]					Goat [38,39,40]		
	Santa Inéz	Merino *	Namaqua	Dorper	Dohne Merino	Saanen *	Tankwa	Boer
**Semen and Sperm Traits**	(*n* = 24)	(*n* = 10)	(*n* = 12)	(*n* = 10)	(*n* = 12)	(*n* = 3)	(*n* = 7)	(*n* = 3)
*Macroscopic*								
Semen pH	−	−	−	−	−	−	−	6.94
Semen volume (mL)	−	−	1.09 ± 0.08	1.37± 0.08	1.2 ± 0.08			0.64 ± 0.11
*Microscopic*								
Total motility (%)	−	74.1 ± 10.7	70–85	70–85	70–85	74.1 ± 10.7	78.0 ± 12.5	78
Sperm concentration (×10^6^/mL)	−	3846.1 ± 2773.0	1220 ± 5200	1100 ± 5290	1140 ± 5200	2306.1 ± 1669	2134.0 ± 1200	3.02
Live sperm (%)	−	−	67.67 ± 1.94	72.82 ± 1.98	68.59 ± 1.94	−	−	89.2
Abnormal morphology (%)	−	−	−	−	−	−	−	4.6
Intact plasma membrane (%)	−	−	−	−	−	−	−	88.2
*Kinematics*								
Average VCL (µm/s)	357 ± 30.7	231.3 ± 48.4	−	−	−	166.0 ± 36.4	157.0 ± 28.7	200.0 ± 14.1
Rapid VCL (µm/s)	−	246.6 ± 38.4	−	−	−	237.3 ± 44.4	225.0 ± 33.4	−
Average VSL (µm/s)	197.8 ± 20.6	182.3 ± 38.6	−	−	−	88.5 ± 33.6	88.9 ± 12.6	117.0 ± 5.94
Rapid VSL (µm/s)	−	188.9 ± 28.4	−	−	−	143.4 ± 28.4	165.0 ± 12.1	−
Average VAP (%)	249.5 ± 23.4	208.3 ± 39.1	−	−	−	104.0 ± 33.1	125.0 ± 18.5	126.0 ± 6.79
Rapid VAP (%)	−	211.9 ± 34.8	−	−	−	153.0 ± 27.8	178.0 ± 19.1	−
Average LIN (%)	55.5 ± 4.8	74.4 ± 12.0	−	−	−	54.3 ± 14.0	70.0 ± 11.1	58.3 ± 1.56
Rapid LIN (%)	−	82.5 ± 5.1	−	−	−	60.6 ± 5.1	73.8 ± 4.5	−
Average STR (%)	79.3 ± 3.5	87.2 ± 5.1	−	−	−	81.8 ± 7.2	87.4 ± 13.2	90.1 ± 0.69
Rapid STR (%)	−	89.3 ± 1.3	−	−	−	93.4 ± 2.4	92.7 ± 12.3	−
Average WOB (%)	−	84.8 ± 10.3	−	−	−	64.0 ± 10.3	79.5 ± 6.3	−
Rapid WOB (%)	−	92.4 ± 5.1	−	−	−	65.0 ± 5.1	79.7 ± 7.1	−
Average ALH (µm)	3.0 ± 0.2	3.6 ± 0.8	−	−	−	3.4 ± 0.7	2.3 ± 0.6	6.1 ± 0.29
Rapid ALH (µm)	2.3 ± 0.2	3.8 ± 0.9	−	−	−	4.5 ± 0.9	3.25 ± 0.6	−
Average BCF (Hz)	53.7 ± 2.0	17.4 ± 2.9	−	−	−	28.8 ± 3.1	32.3 ± 12.1	44.6 ± 0.57
Rapid BCF (Hz)	−	18.1 ± 3.2	−	−	−	28.7 ± 3.2	34.0 ± 15.9	−
Hyperactivation (%)	−	−	−	−	−	−	25.3	−

VCL: curvilinear velocity; VSL: straight-line velocity; VAP: average path velocity; LIN: linearity; STR: straightness; WOB: wobble; ALH: amplitude of lateral head displacement; BCF: beat cross frequency. * Exotic breed farmed in South Africa.

**Table 6 animals-12-00657-t006:** Comparison of semen and sperm functional characteristics of indigenous and crossbred bulls in South Africa to consider similarities in sperm motility parameters. All three crossbreds were between Simmental as the exotic breed and one of the indigenous southern African cattle breeds. This data forms part of a preliminary study where sperm motility assessments were performed by means of computer-aided sperm analysis (CASA) at a high frame rate (169 frames per second), and bulls were randomly selected for measurement (unpublished research).

	Indigenous			Crossbreeds (with Simmental * (S))		
	Nguni (N)	Bonsmara (B)	Afrikaner (A)	S × N	S × B	S × A
**Sperm functionalities**	(*n* = 2)	(*n* = 2)	(*n* = 2)	(*n* = 2)	(*n* = 2)	(*n* = 1)
Total motility (%)	82.5	71	72	85	76	83
Progressive motility (%)	62	55	38.5	76	24.5	62
Rapid progressive motility (%)	11	10.6	8	8	11	12
Rapid progressive VCL (µm/s)	229.1	217	205	219	233	194
Rapid progressive VAP (µm/s)	130.2	108	115.8	98.3	122	108
Rapid Progressive VSL (µm/s)	115	98.2	101.5	85.9	109.5	95
Rapid progressive LIN (%)	50.9	45.1	49.5	39	47.2	49.3
Hyperactivation (%)	41.3	25	37	76	9	32

* Exotic breed farmed in South Africa. VCL: curvilinear velocity; VSL: straight-line velocity; VAP: average path velocity; LIN: linearity.

## Data Availability

We reported data from the literature.

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
