# Peer review of "Origin, Migration, and Reproduction of Indigenous Domestic Animals with Special Reference to Their Sperm Quality"

_animals, 2022, doi:10.3390/ani12050657_

Round 1

Reviewer 1 Report

  • There is data from table 2 and figure 2 that do not match (goat and sheep)

  • Point 4 4. Natural selection vs artificial selection for reproduction, must be summarized, as it is repetitive.

  • Many parameters in the tables presented are incomplete. p. ex. Table 3, 4 and 5, data for volume, mass motility, kinetics, ...

  • In many tables the number of individuals is very low, a low sample and also there are no statistics of the data.

Author Response

Author response: AR

AR: We want to thank the reviewer for the useful comments which assisted to improve the paper.

  • There is data from table 2 and figure 2 that do not match (goat and sheep)

AR: We agree with the discrepancy indicated and made the required changes in Table 2 to match the information provided in Figure 3.

  • Point 4 4. Natural selection vs artificial selection for reproduction, must be summarized, as it is repetitive.

AR: We reviewed this section and eliminated possible repetitions.

  • Many parameters in the tables presented are incomplete. p. ex. Table 3, 4 and 5, data for volume, mass motility, kinetics, ...

AR: Yes, this is indeed the case but we want to remind the reviewer that it is predominantly a review paper where we rely on existing literature and, while we tried to summarize as comprehensively as possible, the gaps in the tables signifies a lack of data in the literature and not inadvertently omitted data. It is simply a case of some data sets providing important information on specific parameters while other parameters have not been investigated or reported in the various research papers. Furthermore, we wanted to include data on more modern sperm functional parameters (related to an individual’s potential fertility), which most papers on sperm characteristics do not include. For example, in Table 3 for Venda and White Leghorn chickens, computer-aided sperm analysis (VCL, VSL and VAP) that objectively quantifies sperm motility, have been performed which has rarely been done in other studies.

  • In many tables the number of individuals is very low, a low sample and also there are no statistics of the data.

AR: Again, most data is extracted from the literature and represents what is available/published and accordingly the relatively low numbers. In Table 6 with only n=2 refers to our own preliminary results. We want to emphasize that this merely represents a random extraction of data to show that most values for most sperm functional parameters fall within the same range. An important aspect relates to showing that indigenous sperm characteristics are at least as good as animals which have been bred for high fertility. Furthermore, it was not really possible to perform statistics among breeds or species in Table 3-5 as we rely on published data in table format and did not have access to raw data. So, the idea is again to show that generally there are large overlaps when the ranges of parameters of indigenous breeds are compared to exotic breeds and accordingly a great likelihood that they may not be significantly different. This aspect was highlighted in the Conclusion.

Reviewer 2 Report

This reviewer would like to congratulate the Authors for this review article, which is considered interesting not only for the scientific community but also from a practical point of view thants to the information gathered. In this R opinion, the manuscript is well-written, easy to read and with a nice thread, easy to follow. 

Some specific comments: there are minor type errors (as in line 11), and some double spaces along the text that this R would recommend to eliminate prior to publication. 

Last, and due to the more biochemist background of this R, I agree with the Authors in the need of considering some other features for determining sperm quality, which would be very interesting for the study of indogenous domestic animals (i.e., omics). Probably, an extended paragraph regarding this issue (or a criticism about the absence/shortage of information) is the only point missing in this manuscript, in this R opinion. 

Author Response

This reviewer would like to congratulate the Authors for this review article, which is considered interesting not only for the scientific community but also from a practical point of view thants to the information gathered. In this R opinion, the manuscript is well-written, easy to read and with a nice thread, easy to follow. 

Author response: AR

AR: We want to thank the reviewer for the positive comments.

Some specific comments: there are minor type errors (as in line 11), and some double spaces along the text that this R would recommend to eliminate prior to publication. 

AR: We have carefully checked for and corrected typos and errors throughout the manuscript.

Last, and due to the more biochemist background of this R, I agree with the Authors in the need of considering some other features for determining sperm quality, which would be very interesting for the study of indigenous domestic animals (i.e., omics). Probably, an extended paragraph regarding this issue (or a criticism about the absence/shortage of information) is the only point missing in this manuscript, in this R opinion. 

AR: We agree that this is an important aspect and added this as a suggestion for future studies to address the lack of information at the end of Section 6. We also included two references as examples of such studies done on human sperm.

Reviewer 3 Report

The purpose of the study (according to the authors) was to focus on the reproductive or fertility potential of indige domestic animals. After reading this paper, I have slightly different feelings. More than half of the text focuses on the historical aspects of domestication, while the chapter on fertility appears to be secondary.

- adapt the abstract to the journal's requirements (structural division is not accepted)

- Keywords; avoid creating keywords with more than one or two words

- the work focuses mainly on animals domesticated in Africa, therefore both the title and the purpose of the work require modification, as they suggest that the authors refer to all domesticated mammals in this text. In addition, it should be clarified which group of mammals is covered by this review.

Lines - 43 to 105 - this part should be removed or reduced to the necessary minimum. It has little to do with the purpose of the work.

Lines 259-329 the authors describe in detail the differences between the breeds and species of the tested animals in terms of semen quality, instead of explaining in detail how the domestification process influenced the formation of these biological quality traits. They subject themselves to slight deliberations on this subject in a short fragment; lines 330-345

Lines 346-348 I have the impression that this is a fragment accidentally copied from a previous review of this work

I also did not find a significant chapter on females, their fertility in the context of domestication, or conclusions drawn from the current state of knowledge. Moreover, in my opinion, the work should contain a comprehensive and tabular summary of all factors influencing the fertility of the discussed animals, changes in fertility over several decades, and all this should be preceded by a detailed discussion of the topic.

Author Response

Author Response = AR

AR: We want to thank the reviewer for the constructive criticism which assisted greatly to improve the quality of the paper.

The purpose of the study (according to the authors) was to focus on the reproductive or fertility potential of indige domestic animals. After reading this paper, I have slightly different feelings. More than half of the text focuses on the historical aspects of domestication, while the chapter on fertility appears to be secondary.

AR: We agree with some of the above comments particularly those related to the historical components. We need to point out that in our title “Origin” clearly signifies the importance of the history and we now added “Migration” to the title since we believe it is actually by viewing both the origin and particulalrly the migration of indigenous livestock that natural selection as an important selective force is introduced.  This then provides a more understandable exposition of the relationships of the migration facets and sperm quality. Please note you stated fertility appears to be secondary. We did not address fertility per se but rather semen and sperm functionality. By implication it would infer potential fertility but one simply does not know.

- adapt the abstract to the journal's requirements (structural division is not accepted)

AR: We have adapted the format of our abstract to the journal’s requirements.

- Keywords; avoid creating keywords with more than one or two words

AR: We have kept to single keywords as far as possible but retained phrases such as “natural selection”.

- the work focuses mainly on animals domesticated in Africa, therefore both the title and the purpose of the work require modification, as they suggest that the authors refer to all domesticated mammals in this text. In addition, it should be clarified which group of mammals is covered by this review.

AR: We are not entirely in agreement with the above comment. We have shown that the African livestock are all derived from the fertile crescent (Middle East) and the same applies to most livestock arriving and being domesticated elsewhere in the world. We therefore used Africa as an example to explain some important principles and now highlight this fact (see beginning of section 3). We also allude to other parts of the world in both our tables and figures. A review of domestication of animals globally is outside the scope of this paper. Furthermore, we made it clear that we are dealing largely with most groups of domestic animals, commonly known as livestock (cattle, pigs, goats, sheep, chicken as the predominant ones) and indicated it clearly in the Introduction and in several of our tables. Thus, we did not focus on mammals per se and removed this as one of the keywords.

Lines - 43 to 105 - this part should be removed or reduced to the necessary minimum. It has little to do with the purpose of the work.

AR: The mentioned section essentially forms the background of the review showing the origins of domestication and accordingly the subsequent migrations, as is indicated in the changed title. Since we have adopted an evolutionary approach, more in-depth explanations regarding the wild progenitors and period in history when domestication started contributes in the better understanding of the resilience of indigenous livestock and why it would be important to preserve the genes that make them so hardy.

Lines 259-329 the authors describe in detail the differences between the breeds and species of the tested animals in terms of semen quality, instead of explaining in detail how the domestication process influenced the formation of these biological quality traits. They subject themselves to slight deliberations on this subject in a short fragment; lines 330-345

AR: Thank you for these insights into our work. We still maintain that it is the migration through harsh environments and natural selection that must have selected for survival/reproduction and accordingly good sperm traits. It is the Darwinian theme of natural selection (adapt to your environment or die) which firstly imply selection to reproduce. Subsequently, during the domestication process, animals with high fertility or good sperm traits were selected (artificially) for breeding and thus maintaining these traits. In some instances, the literature indicates that artificial selection may not be favourable, e.g. crossbreeding. The authors are of the opinion that we have addressed these aspects in the manuscript.

Lines 346-348 I have the impression that this is a fragment accidentally copied from a previous review of this work

AR: Thank you for this comment – we have omitted this fragment.

I also did not find a significant chapter on females, their fertility in the context of domestication, or conclusions drawn from the current state of knowledge. Moreover, in my opinion, the work should contain a comprehensive and tabular summary of all factors influencing the fertility of the discussed animals, changes in fertility over several decades, and all this should be preceded by a detailed discussion of the topic.

AR: The title speaks about reproduction but clearly with a focus on males and spermatology. Accordingly, while female reproduction is equally important it was only a peripheral aspect of our paper and we now emphasized this in the Introduction that it is regarded as peripheral for the purpose of this paper. We have not dealt with fertility as such and information on fertility in livestock over several decades is essentially not described in the literature. Historically, semen analysis techniques were all manual and open to great variation in assessment and would make comparisons difficult. We also believe that even if information was available over several decades, it is futile in the context of thousands and hundreds years of natural selection moulding these characteristics. Furthermore, this is why we have spent quite some discussion time on the issue of natural versus artificial selection and how that moulded current livestock populations in terms of their reproductive ability with reference to spermatology.

Round 2

Reviewer 1 Report

The modifications made to the original document have been few. The article goes on to present shortcomings, such as the fact that the number of animals is not very significant, and that tries to relate the seminal quality with factors, with which it is really difficult to have any relationship. And the data for all species and races are not presented equally.

Author Response

Reviewer comment: The modifications made to the original document have been few. The article goes on to present shortcomings, such as the fact that the number of animals is not very significant, and that tries to relate the seminal quality with factors, with which it is really difficult to have any relationship. And the data for all species and races are not presented equally

Author response: We have responded in the previous version to each and every comment and explained in detail why there are omissions in the Tables and that related to small numbers and we merely tried to make the point that even using a random selection there are no differences in semen traits among indigenous and exotic livestock. 

Reviewer 3 Report

The authors explained all the objections raised to the manuscript in their responses. The text has been refined enough.

Author Response

We appreciate the positive comments of the Reviewer that we made the required changes. We also did a minor spell check as requested